



# KLT-IV v1.0: Image velocimetry software for use with fixed and mobile platforms

Matthew. T. Perks[1]

[1]School of Geography, Politics and Sociology, Newcastle University, Newcastle upon Tyne, United Kingdom.

**Abstract.** Accurately monitoring river flows can be challenging, particularly under high-flow conditions. In recent years, there has been considerable development of remote sensing techniques for the determination of river flow dynamics. Image velocimetry is one particular approach which has been shown to accurately reconstruct surface velocities under a range of hydro-geomorphic conditions. Building on these advances, a new software package, KLT-IV v1.0 has been designed to offer

a user-friendly graphical interface for the determination of river flow velocity and river discharge using videos acquired from a variety of fixed and mobile platforms. Platform movement can be accounted for when ground control points and/or stable features are present, or where the platform is equipped with a differential GPS device and inertial measurement unit (IMU) sensor. The application of KLT-IV v1.0 is demonstrated using two case studies at sites in the UK: (i) River Feshie; and (ii) River Coquet. At these sites, footage is acquired from unmanned aerial systems (UAS) and fixed cameras. Using a combination

of ground control points (GCPs), and differential GPS and IMU data to account for platform movement, image coordinates are converted to real world distances and displacements. Flow measurements made with a UAS and fixed camera are used to generate a well-defined flow rating curve for the River Feshie. Concurrent measurements made by UAS and fixed camera are shown to deviate by $< 4\%$ under high-flow conditions where maximum velocities exceed $3\,\mathrm{m\,s^{-1}}$. The acquisition of footage on the River Coquet using a UAS equipped with differential GPS and IMU sensors enabled flow velocities to be precisely

reconstructed along a $180\,\mathrm{m}$ river reach. In-channel velocities of between 0.2 and $1\,\mathrm{m\,s^{-1}}$ are produced. Check points indicated that unaccounted for motion in the UAS platform is in the region of $6\,\mathrm{cm}$. These examples are provided to illustrate the potential for KLT-IV to be used for quantifying flow rates using videos collected from fixed, or mobile camera systems.

## 1 Introduction

### 1.1 Challenges in hydrometry

Observed flow rates in rivers represent the integration of water basin input, storage and water transfer processes. Accurate long-term records are essential to understand variability in hydrological processes such as the rainfall-runoff response (Hannah et al., 2011; Borga et al., 2011). This information provides the foundation for accurate predictions of hydrological response to catchment perturbations, and is the basis of informed water resources planning and the production of effective catchment-based management plans.





Current approaches for the quantification of river flow are generally applied at strategic locations along river networks through the installation of fixed monitoring stations. Many of these stations are reliant on the development of an empirical stage-discharge rating curve, which is often achieved by developing an empirical function between paired measurements of river flow (combining measurements of velocity and cross-section area) and river stage measurements. This empirical function is then applied to a continuous record of stage measurements to predict flow discharge (Coxon et al., 2015). Obtaining accurate

flow gaugings using traditional approaches can be challenging, often costly and time consuming, with flow observations during flood conditions being hazardous to operatives. Resultantly, considerable progress being made in the development of remotely operated fixed and mobile systems capable of providing quantitative estimates of instantaneous and time-averaged flow characteristics. Examples of successful developments include acoustic doppler current profilers (Despax et al., 2019) and microwave radar sensors (Welber et al., 2016). Whilst advances in technology have led to more accurate, and safer flow gaugings in some

areas, these devices can be costly thereby limiting their adoption to locations of high priority. In contrast to the investment required to implement these new techniques and technologies, continued funding and resource pressures faced by competent authorities in Europe and North America has led to a decline in investment in recent years, with reductions in the number of monitoring stations (Stokstad, 1999). This poses a real threat to the continuity of river flow data archives and has the potential to compromise our ability to detect future hydrological change.

As a consequence, innovative solutions are required to reduce the cost and time-intensive nature of generating river discharge data in order to ensure the long-term sustainability of hydrometric infrastructure and hydrological records. With the development and implementation of new solutions, improvements in monitoring from ground-based and remotely operated platforms may ensue, with hydrometric monitoring networks becoming tailored to meet the demands of modern water resources management (Cosgrove and Loucks, 2015).

### 1.2 Aim


Taking into consideration the aforementioned challenges to monitoring hydrological processes, KLT-IV aims to provide the user with an easy to use, graphical interface for the determination of flow rates using videos acquired from fixed, or mobile platforms. In this article, the following sections are presented: (i) an overview of existing image-based hydrometric solutions; (ii) details of the underlying methodology of KLT-IV and the features that are supported; (iii) examples demonstrating several

KLT-IV work-flows including the associated outputs generated by the software; and (iv) perspectives on the challenges relating to further development of image velocimetry software.

### 1.3 Image-based hydrometric solutions: Existing work-flows and limitations

Amongst the recently developed approaches offering a great deal of promise for monitoring surface flows is image velocimetry. The fundamental basis of the image velocimetry approach to flow gauging is that the detection and subsequent rate at which

optically visible, or thermally distinct surface features, e.g. materials floating on the water surface (foam, seeds, etc.) and water surface patterns (ripples, turbulent structures), are displaced downstream can be used to estimate the surface velocity of the water-body. The surface velocity may then be converted to a depth-averaged velocity by fitting a power or logarithmic law to





vertical velocity profile observations (Welber et al., 2016), or this may be theoretically derived assuming a logarithmic velocity profile (Wilcock, 1996). Image velocimetry is an innovative solution for measuring stream-wise velocities, understanding flow

patterns and hydrodynamic features. This information can later be supplemented with topographic and bathymetric observations to determine the discharge of surface water-bodies.

The first step in any large-scale image velocimetry work-flow is obtaining image sequences for subsequent analysis. Due to technological advancements, this is commonly achieved through the recording of videos in high-definition and at a consistent frame rate. Camera sensors also have a range of sizes and focal lengths, which offers the opportunity for the choice of

instrument to be defined based on the conditions of operation (e.g. distance to area of interest, required angle of view, etc.). Following video capture, images are extracted for subsequent analysis along with meta-data (e.g. video duration, frame rate, number of frames).

Following image acquisition, image pre-processing can be performed to alter the color properties of the images. Example operations include histogram equalization, contrast stretching, application of a high-pass filter, and binariazation. Image pre-

processing is usually applied to enhance the visibility of surface water features against the background, eliminate the presence of the river-bed, or to reduce glare. These options are present within some existing image velocimetry software packages (e.g., Thielicke and Stamhuis, 2014), and also open source image processing software packages (e.g., ImageJ, Fiji, 2020; Schindelin et al., 2012).

Following image enhancement, the choice of image-pairs used to determine displacement needs to be carefully considered

in most work-flows and this is a function of the sensor resolution, acquisition frame rate, ground sampling distance, and flow conditions (Legleiter and Kinzel, 2020). Image pairings must be selected to ensure that the displacement of surface features is sufficient to be captured by the sensor, but short enough to minimise the potential for surface structures to transform, degrade, or disappear altogether, or for external factors to influence the measurement (e.g. camera movement in the case of unmanned aerial system (UAS) deployments; Lewis and Rhoads, 2018). Therefore, the optimum image sampling rate needs to be established

on a case-by-case basis and requires the operator to have a level of experience and expertise (Meselhe et al., 2004). These considerations also feed into the selection of an appropriate size of interrogation area (or equivalent). This needs to be large enough for sufficient surface features to form a coherent pattern for cross-correlation algorithms to be applied. However as this area increases, so does the the uncertainty in valid vector detection as a result of the size the correlation peak decreasing (Raffel et al., 2018). Whilst recommendations have been made over the determination of these settings (e.g., Raffel et al.,

2018; Pearce et al., 2020), they have the potential to significantly alter the quality of the velocity computations. The recent application of multiple-pass, or ensemble correlation approaches, has however been shown to improve the analysis accuracy and the production of results in closer agreement better agreement to reference values than single-pass approaches (Strelnikova et al., 2020).

Prior to the application of image velocimetry algorithms, a series of image processing steps may be required. In some cases,

small-scale vibrations (e.g. by wind, traffic) can result in random movement of a fixed camera, or alternatively, if the camera is attached to a UAS or helicopter the camera may drift over time (Lewis and Rhoads, 2018). This can result in image sequences that are not stable with apparent ground movement in parts of the image where there is none. Many image velocimetry software





packages currently neglect this stage, with the notable exception of Fudaa, RiVER, and FlowVeloTool. These software packages are able to account for small amounts of camera movement through the application of projective or similarity transformations

based on automated selection and tracking of features within stable parts of the image. However, these approaches require significant elements within the image field-of-view to be static, which may not always be possible. Furthermore, this approach does not allow for the complete translation of a scene (e.g., Detert et al., 2017).

Upon the compilation of a sequence of stabilised images, pixel coordinates of the image are usually scaled to represent real-world distance. This can be applied using a direct scaling function where the relationship between pixel and metric coor-

dinates is already known, and is stable across the image and throughout the image sequence (i.e. the lens is rectilinear (or the distortion has been removed), lens is positioned orthogonal to the water surface and stable). Alternatively, in instances where these assumptions do not hold true, image orthorectification can be conducted. In this process, ground control points (GCPs) may be used to establish the conversion coefficients, which are then used to transform the images. In this approach the transformation matrix implicitly incorporates both the external camera parameters (e.g. camera perspective), and the internal camera

parameters (e.g. focal length, sensor size, and lens distortion coefficients). Where ground control points are located planar to the water surface, a minimum of four GCPs is required (Fujita et al., 1998; Fujita and Kunita, 2011), or in the case of a three-dimensional plan-to-plan perspective projection a minimum of six GCPs distributed across the region of interest, are required for the determination of orthorectification coefficients (Jodeau et al., 2008; Muste et al., 2008). Alternatively, the sources of image distortion may be explicitly modelled (e.g., Heikkilä and Silvén, 2014), enabling intrinsic parameters to be determined

through calibration, and applied to alternative scenes (e.g., Perks et al., 2016). This reduces the dependency on ground control points provided that intrinsic parameters are known and optimisation is limited to the external camera parameters (i.e. camera location, view direction). More recently, the integration of supplementary sensors (e.g. differential GPS, inertial measurement unit (IMU)), and associated measurements for determining orthorectification parameters has been advocated for (e.g., Legleiter and Kinzel, 2020), but this approach has yet to be embedded into image velocimetry software.

Upon the determination, or optimisation of the transformation matrix, which is used to map pixel coordinates to ground coordinates, there are two divergent approaches of how to use this information to generate velocity information in real-world distances. The most widely used approach is to use the transformation coefficients to generate a new sequence of images where ground distances are equivalent across all pixels across the image. Image velocimetry analysis is then conducted on this orthorectified imagery. However, some work-flows neglect this stage, instead conducting image velocimetry analysis on the

raw images, and applying a vector correction factor to the velocity vectors (e.g., Fujita and Kunita, 2011; Perks et al., 2016). The benefit of the latter approach is that image velocimetry analysis is conducted on footage that has not been manipulated or transformed and therefore there is no opportunity for image processing artefacts to influence the velocity outputs. Conversely, an advantage of direct image transformation is that the parameters being applied by the image velocimetry algorithms are consistent throughout the image (e.g. 32 x 32 px represents the same ground sampling area across the entirety of the image).

Following image pre-processing, stabilisation, and orthorectification (when required), a range of image velocimetry approaches may be used to detect motion of the free-surface. Large-scale Particle Image Velocimetry (LSPIV) is built upon the Particle Image Velocimetry (PIV) approach commonly employed in laboratory settings. This approach applies two-dimensional





cross correlation between image-pairs to determine motion. The first image is broken into cells (search areas) within a grid of pre-defined dimensions and these search areas are used as the template for the two-dimensional cross correlation. In the
second image, an area around each search area is defined and the highest value in the two-dimensional cross-correlation plane is extracted and is used as an estimate of fluid movement. Space–time Image Velocimetry (STIV) was inspired by LSPIV and searches for gradients between sequences of images by stacking sequential frames and searching for linear patterns of image intensity (Fujita et al., 2007, 2019). Similarly to PIV, Particle Tracking Velocimetry (PTV) can also be based on cross-correlation, but rather than utilising an aggregation of surface features (patterns) to determine movement, individual surface
features are selected, and their likely displacement determined. Upon acquisition of displacement estimates, post-processing in the form of vector filtering can be applied. This may take the form of correlation thresholds, manual vector removal, a standard deviation filter, local median filter (Westerweel and Scarano, 2005; Strelnikova et al., 2020), trajectory-based filtering (Tauro et al., 2019), or imposing limits to the velocity detection thresholds.

A final element in image velocimetry work-flows for the determination of river discharge involves the incorporation of ex-
ternal data. Firstly, the free-surface image velocity measurements must be translated into a depth-averaged velocity. Buchanan and Somers (1969) and Creutin et al. (2003) provided estimates of $0.85 - 0.90$ as an adequate ratio between surface and depth-averaged velocities under the condition of a logarithmic profile. This has been found to hold true for a number of environmental conditions (e.g., Le Coz et al., 2007; Kim et al., 2008), with maximal deviations from these default values of less than 10% (Le Coz et al., 2010). However, this should ideally be informed by direct measurements made in the area of interest. It may also
be the case that determining the displacement across the entire cross-section is not possible (e.g. due to lack of visible surface features). Therefore, interpolation and extrapolation may need to be undertaken. This may be achieved using the assumption that the Froude number varies linearly or is constant within a cross-section (Le Coz et al., 2010), or based on theoretical flow field distributions (Leitão et al., 2018). Upon a complete profile, unit discharge can be calculated based on the specified water depth at a number of locations in the cross-section and this is then aggregated to provide the total river flow.
Building on the existing image velocimetry software packages that are currently available, and seeking to address some of their limitations, KLT-IV v1.0 seeks to offer a novel, flexible approach to acquiring hydrometric data using image-based techniques. The specific details of this approach are introduced in Section 2.

## 2  Methods

### 2.1  Software Background

A new branch of PTV has recently been explored, whereby features are detected based on two-dimensional gradients in pixel intensity across the image using one of a range of automated corner point detection algorithms (e.g. SIFT, GFTT, FAST). These features are subsequently tracked from frame to frame using optical flow techniques. This approach has only recently been used for the characterisation of hydrological processes with examples including monitoring of a fluvial flash flood using a UAS (Perks et al., 2016), application of optical tracking velocimetry (OTV) on the Tiber and Brenta rivers using fixed gauge
cams (Tauro et al., 2018), benchmarking exercises using a range of image velocimetry approaches including KLT-IV and OTV





(Pearce et al., 2020), and in the development of FlowVeloTool (Eltner et al., 2020). Optical flow-based approaches have the benefit of being computationally efficient whilst being capable of automatically extracting and tracking many thousands of visible features within the field of view.

The underlying approach of KLT-IV is the detection of features using the Good Features To Track (GFTT) algorithm (Shi
and Tomasi, 1994), and subsequent tracking using the pyramidal Kanade Lucas Tomasi tracking scheme (Lucas et al., 1981; Tomasi and Kanade, 1991). The three level pyramid scheme allows for a degree of flexibility in the user specified interrogation area (block size). The interrogation area is refined between pyramid levels by down-sampling the width and height of the interrogation areas of the previous level by a factor of two. An initial solution is found for the lowest resolution level and this is then propagated through to the highest resolution. This enables features to be tracked that extend beyond the initial interrogation
area. A total of thirty search iterations are completed for the new location of each point until convergence. Error detection is established by calculating the bidirectional error in the feature tracking (Kalal et al., 2010). If the difference between the forward and backward tracking between image pairs produces values that differ by more than 1 px, the feature is discarded. Depending on the configuration adopted (see Section 2.2.2), feature tracking is conducted on either orthorectified imagery with the resultant feature displacement in metric units, or on the raw imagery with vector scaling occurring after analysis. When
pixel size is not explicitly known in advance, the transformation between pixel and physical coordinates is achieved through the generation and optimisation of a distorted camera model (Messerli and Grinsted, 2015; Perks et al., 2016), which in the case of moving platforms, is updated iteratively based upon GCPs, or differential GPS data. KLT-IV is a standalone graphical user interface (GUI) developed in MATLAB 2019, with the incorporation of Cascading Style Sheets (CSS) to enable flexibility of interface design (StackOverflowMATLABchat, 2016). The application is compiled as a standalone executable and is packaged
with ffmpeg (version N-93726-g7eba264513) (see Software Availability).

## 2.2 Interface

The interface is split up into five sections: (i) Video Inputs; (ii) Settings; (iii) Ground Control; (iv) Analysis; and (v) Discharge (Figure 1). Within each of these categories are a number of options which automatically activate and deactivate depending on the type of orientation selected. Consequently, there are a number of potential work-flows, and these will be outlined in this
section. All inputs are in units of meters unless specified otherwise.

### 2.2.1 Video Inputs

The first section: Video Inputs, is where the video acquisition details are provided. The default mode in this version is 'Single Video', meaning that only one video at a time can be analysed, and this video may be selected using the file selection dialog box. There is flexibility in the video formats that may be used within the software as outlined in Appendix A. Upon selecting
a video, the user is provided with the option to re-encode the footage. Under most instances this is not required. However, on some occasions Internet Protocol (IP) cameras may fail to embed the correct meta-data (e.g. image resolution, frame rate) within the video. Accurate meta-data are an essential pre-requisite to accurate analysis and re-encoding the video can restore this information. If re-encoding is selected this process is automatically undertaken using using the libx264 encoder. This





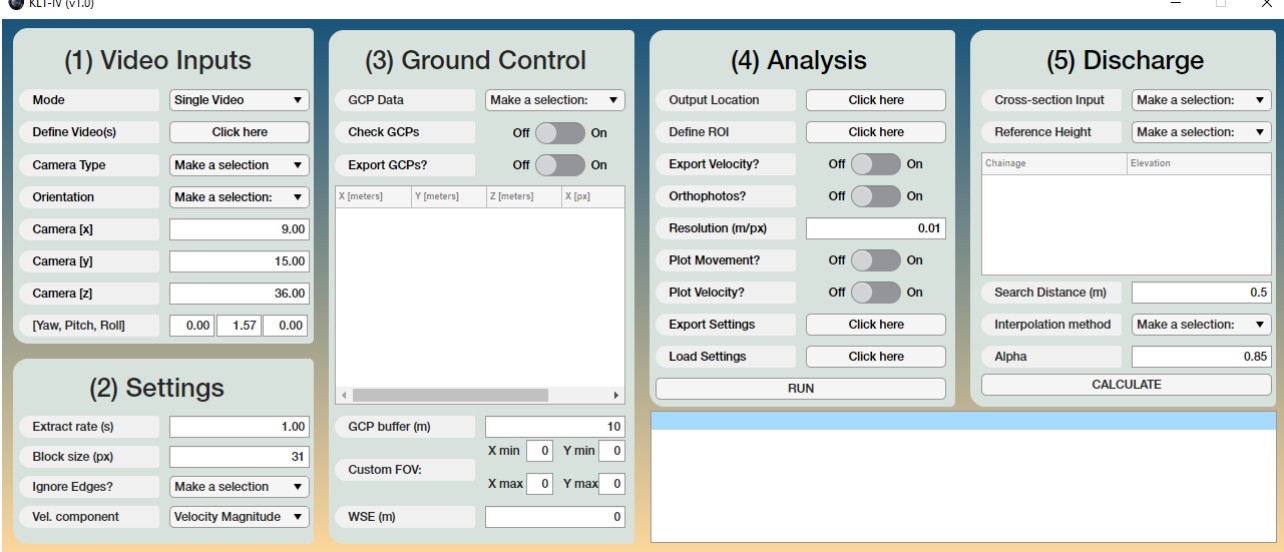

**Figure 1.** Graphical User Interface (GUI) of KLT-IV version 1.0

results in the generation of a new video within the same folder as the input with the suffix '_KLT' appended to the input file
name.

The next option allows the user to specify the camera type used to acquire the footage. A number of IP, hand-held, and UAS-mounted cameras are available for selection. If the user selects a camera, the calibrated internal camera parameters will be used during the orthorectification process. This enables fewer parameters to be solved for in the optimisation process. However, if the 'Not listed' option is chosen then the internal camera parameters are optimised during orthorectification process. If

orientation [A] or [F] is selected (Table 1), the camera model is a required input, otherwise it is optional. The camera models used in KLT-IV been developed through the use of a checkerboard pattern and the Camera Calibrator App within MATLAB.

Next, one of six camera orientations can be chosen and these are outlined in Table 1. Each of the chosen options has different input requirements from this point forwards (Figure 2). Some work-flows require the input of the camera location [X, Y, Z] and camera view direction [yaw, pitch, roll]. If the orientation uses GCPs, this positioning information is used as the starting

point for the development of the camera model. If 'Stationary: Nadir' is selected as the orientation, the camera [z] coordinate should be precisely defined relative to the water surface elevation (see Section 2.2.3), as the distance between camera and water surface is used to define the conversion between pixel size and metric coordinates. The camera yaw, pitch, and roll settings should be provided in radians. In the case of the yaw angle, 0 equates to East, 1.57 equates to North, 3.14 equates to West, and 4.71 equates to South. These bearings are provided relative to the GCP coordinate system. A pitch of 1.57 equates to the

camera oriented at nadir with each degree of inclination subtracting 0.017 from this value. Generally, camera roll is negligible and the default value (zero) can be adopted.



**Table 1.** A summary of the assumptions, requirements, advantages, and limitations of the different orientation options found with Section 1: Video Inputs of KLT-IV v1.0.

| Orientation | Assumptions | Requirements | Advantages | Limitations |
|---|---|---|---|---|
| Stationary: Nadir [A] | The camera is stationary and view is nadir | Defined camera location and camera model, camera oriented at nadir, known water surface elevation | No GCPs required | Assumption of stable and nadir camera |
| Stationary: GCPs [B] | The camera is stationary and GCPs are present | Estimated camera location, estimated view direction, GCPs, water surface elevation | Camera calibration leading to accurate trajectories | Assumption of stable camera |
| Dynamic: GCPs [C] | The camera may be mobile and GCPs are present | Estimated camera starting location, estimated view direction, GCPs, water surface elevation | Camera calibration using GCPs tracked between frames, scene can be dynamic with addition of GCPs over time | GCPs should be clearly visible |
| Dynamic: GCPs & Stabilisation [D] | The camera may be mobile, and GCPs are present but may be difficult to track | Estimated camera starting location, estimated view direction, GCPs, water surface elevation | Frames are stabilised relative to the first frame to account for movement, GCPs do not need to be clearly visible | Assumption that the area outside of the defined ROI is stable, camera perspective (i.e. pitch) does not alter significantly, bank-side features are at similar elevations |
| Dynamic: Stabilisation [E] | The camera may be mobile, GCPs are not present but pixel size is known | Pixel scaling (px/m), water surface elevation | Frames are stabilised relative to the first frame to account for movement, GCPs are not required | Assumption that pixel scaling is constant across image, the area outside of the defined ROI is stable, camera perspective does not alter significantly |
| Dynamic: GPS & IMU [F] | The camera may be mobile, differential GPS and IMU data are used to define the camera model and sequential images are stabilised | High rate PPK/RTK GPS and IMU data, water surface elevation, camera at nadir | No GCPs are required and the platform can be mobile | Precision is dependent on GPS and IMU quality and sample rate, stable features must be visible |





### 2.2.2 Settings

The Settings section provides the user with the opportunity to customise the settings used in the feature tracking process and therefore the determination of velocities. The feature tracking procedure is designed to identify and track visible features between each frame of the video. However, the user may define the length of time that features are tracked for before their displacement is calculated. If for example, the extract rate is defined as $1\,\mathrm{s}$ and the video frame rate is $20\,\mathrm{Hz}$, features would be detected in frame one, and tracked until frame 21, at which point, the displacement of the features are stored. The sequence would then be restarted at frame 21 and continue until 41, etc. The smaller the value given as the extraction rate, the greater the number of trajectories that will be produced, and any areas of unsteady flow elements will be well characterised. However, small feature displacements can be adversely affected by residual camera movement. Higher extract rates provide a smoothing of the trajectories, averaging particle motion over a greater distance. This makes the process more robust and greater confidence can be placed on the resultant values. However, trajectory numbers will be reduced, and a higher degree of spatial averaging will occur. In most instances, values of between 0.5 and $2\,\mathrm{s}$ are generally appropriate. The block size determines the interrogation area during the feature tracking process. As KLT-IV employs a pyramidal scheme, and tracks features frame-by-frame, analysis is relatively insensitive to this value provided the frame rate is sufficiently high (i.e.<5 fps), and pixel ground sampling distance in the order of decimetres or less. The minimum block size value is 5 px and a default value of 31 px proves sufficient for most deployments. During the determination of features to track, features present close to the edges of the video (outer $10\,\%$) can either be ignored or included in the analysis. If using a camera with significant levels of distortion (e.g. DJI Phantom 2 Vision+), it is recommended that the edges are ignored as residual distortion may persist thereby negatively affect the results (Perks et al., 2016). In the present version of the software the velocity magnitude is provided in $\mathrm{m\,s^{-1}}$, along with the X and Y components as defined by spatial orientation of the GCPs.

### 2.2.3 Ground Control

Ground control points may be used to transform the information within the imagery from pixel scale to metric scale i.e. to establish how distances between pixels relate to real-world distance. To achieve this, the physical location [X, Y, Z] of ground control points (GCPs) within the image are required. The locations of the GCPs can be input in one of several ways. If the pixel coordinates are known, these can be manually input into the table within the GUI, ensuring that pixel indices are appropriately referenced with [0, 0] corresponding to the upper left corner of the image. Alternatively, if the data are already saved in a spreadsheet (.csv format), this can be loaded directly using the file selection dialog box. The format should match that of the GUI table (including headers). Finally, if the pixel locations are not yet known, these can be selected directly from the image. If 'Dynamic: GCPs' is selected as the orientation, GCPs are tracked iteratively between frames. If GCPs are difficult to visually identify (and therefore difficult to track), it may be beneficial to enable the 'Check GCPs' option. This enables the user to manually check the location of the GCPs and offers the option to add additional GCPs should they come into view during the video. The GCP data can also be exported as a .csv file for easy import in future. It is recommended that a minimum of 6 GCPs are defined in this process. Next the user defines the spatial extent (field-of-view; FOV) of the images. This can either be




**Figure 2.** Workflow for Section 1 - 3 of KLT-IV v1.0. Different work-flow scenarios are provided based on the choice of camera orientation. Letters correspond to the orientation defined in Table 1. Grey, yellow, and green colors relate to items within the: (1) Video Inputs, (2) Settings, and (3) Ground Control sections respectively. Notes: Either the GCP buffer [1], or Custom FOV [2] should be provided.





defined as a buffer around the initial GCPs or can be defined explicitly. For example, if a GCP buffer of $10\,\mathrm{m}$ is used (default),
        orthophotos will be generated that extend $10\,\mathrm{m}$ beyond the GCP network. Conversely, if a custom FOV is defined, orthophotos
        will be generated for this specified area. This input is required even if orthophotos are not exported (see Section 2.2.4). Finally,
        the user is required to provide the Water Surface Elevation (WSE) in meters. This should be provided in the same coordinate
        system as the camera and the GCPs. For example, if the camera is located at an elevation of $10\,\mathrm{m}$ and the imaged water surface
is located $7\,\mathrm{m}$ below, the water surface elevation would be defined as $3\,\mathrm{m}$.

### 2.2.4   Analysis

The configuration of the outputs is specified in the 'Analysis' section (Figure 3). The location where the outputs are to be
stored is defined using the pop-up dialog box. The region-of-interest (ROI) is manually provided by drawing a polygon around
the area which defines areas within the image where velocity measurements will be calculated. Features tracked outside of
the ROI are not stored during the analysis. This is an optional input and if this is not provided then the extent will match the
area defined by the GCP buffer, or Custom FOV, as specified in Ground Control. The exception to this is if 'Dynamic GCPs +
Stabilisation', or 'Dynamic Stabilisation' is selected, in which case the ROI is required. For these two configurations, the area
defined as being outside of the ROI (outside of the polygon) is used to stabilise the image sequence. It is therefore important
that there is no actual movement outside of the polygon when using these configurations. There is an option to export the
velocities of the tracked particles as a .csv file. Orthophotos may also be generated for the frames at the beginning and end
of the tracking sequence. The user can define the resolution of the orthophotos that are generated (upto a maximum of 180
million cells, equivalent to an area of $134 \times 134\,\mathrm{m}^2$ at a resolution of 0.01 m/px). If this area is exceeded, the resolution will
automatically be scaled by a factor of two (or multiples thereof) until below this threshold. The user can also specify whether
they wish to visualise the estimated movement of the platform (when 'Dynamic: GCPs' is selected), and whether they would
like to plot the particle trajectories. Finally, it is possible to export and load the application settings for future use and these are
saved to the Output Location.

Upon selecting 'RUN', the analysis begins. Firstly, in the case of configurations using GCPs, a camera model is created and
optimised using the GCP information provided. An RMSE of the GCP re-projection error is provided along with a visualisation
of the precision of the orthorectification process. If the solution is poorly defined the user may halt the process at this stage
and provide inputs that better describe the camera [X, Y, Z, view direction] and/or GCPs before re-running. The user is also
provided with the opportunity to limit the analysis to a specific number of seconds of the video. Processing is undertaken on
the video, and updates on the progress are provided within the GUI. A complete overview of the processes undertaken for
each configuration is provided in (Figure 3). Any exports that the user chooses will be saved in the defined output location.
Orthophotos are exported as greyscale .jpg at the defined pixel resolution, and velocity outputs are exported as a .csv file. The
velocity output includes the starting location of tracking (X, Y), the velocity magnitude, and the X and Y flow components
which are within the same orientation as the GCP survey. The estimated movement [X, Y, Z] of the platform is also shown if
selected. Successfully tracked features, and their trajectories are displayed within the specified ROI and the user may choose
how many features to plot. For an overview, 10,000 features is usually sufficient, but this may be increased to 100,000+ if more



**Figure 3.** Work-flow for Section 4 of KLT-IV (shown in blue), and an outline of the image processing routine used in the determination of velocity magnitudes. Capitalised letters in square brackets correspond to the orientation defined in Table 1. Dashed icons represent optional inputs/outputs, which are dependent on the settings provided in Sections 1-4. Red icons represent user inputs which are prompted once the 'Run' button has been pushed.





detail is required. However, as the number of features selected to display increases, so does the demand on the PCs memory
(RAM). Following successful completion of the analysis and export of the selected outputs, the user may continue through to
Section 4 and determine the river discharge.

### 2.2.5  Discharge

The input of a known cross-section is required in order to compute the river discharge. This can be provided in one of two ways.
Firstly, if the cross-section data has the same spatial reference as the camera location/GCP data then a 'Referenced survey' can
be selected. This method enables the user to input the known locations of each survey point in the cross-section. This is most
likely to be appropriate when the same method is used for surveying the GCPs and cross-section (e.g. survey conducted using
a differential GPS device exported into a local coordinate system). Secondly, if measurements of the cross-section were made
at known intervals from a known starting and finishing position that can be identified from within the video footage, the option
'Relative distances' may be selected. In selecting the latter option, the first frame of the video is displayed, and the user is
instructed to choose the start and stop of the surveyed cross-section. Next, the user may define the survey data as being either:
(i) true bed elevation; or (ii) water depth. In the former, the actual bed elevation is provided, whereas in the latter the absolute
water depth is provided. The user is then instructed to load the .csv file containing the survey data. In the case of a referenced
survey, the columns should be [X, Y, Z/Depth] (including a header in the first row), whereas in the case of relative distances
the .csv should be in the format [Chainage, Z/Depth]. Each measurement along the transect is treated as a node for which a
paired velocity measurement is assigned. The user provides a 'Search Distance' which is a search radius around each node.
Using the velocities found within this search radius, the median is stored. In parts of the channel where no features are tracked,
or visible, it may be necessary to interpolate between, or extrapolate beyond measurements. This can be achieved in one of
three ways: (i) quadratic (second order) polynomial - This works well where peak velocities occur in the centre of the channel
and decrease symmetrically towards both banks; (ii) cubic (third order) polynomial - This works well where flow distribution
is asymmetrical or secondary peaks are present; (iii) constant Froude method - The Froude number (Fr $= V/\sqrt{gD}$) (Le Coz
et al., 2008; Fulford and Sauer, 1986) is calculated for each velocity and depth pairing, with this function being used to predict
velocities in areas where no features are tracked. This approach may be particularly beneficial when the flow distribution does
not conform to (i), or (ii).

### 2.3  Case Study Descriptions

Within the following sections, descriptions are provided of two example case studies where footage has been acquired for image
velocimetry purposes. These field sites are located in the UK with footage acquired during low- and high-flow conditions using
fixed cameras and mobile platforms (UAS).





### 2.3.1 Case Study 1: River Feshie, Scotland

The River Feshie, in the Highlands of Scotland, is one of the most geomorphologically active rivers in the UK. The headwaters
originate in the Cairngorm National Park at an elevation of $1263 \, \mathrm{mAOD}$ before joining the River Spey at an elevation of
$220 \, \mathrm{mAOD}$. Approximately $1 \, \mathrm{km}$ upstream of this confluence is a Scottish Environmental Protection Agency (SEPA) gauging
station (Feshie Bridge). This monitoring station at the outlet of the $231 \, \mathrm{km}^2$ Feshie catchment is a critical, but challenging
location for the measurement of river flows. The channel is liable to scour and fill during high-flow events and the natural
control is prone to movement in moderate to extreme spates.

At this location, a Hikvision DS-2CD2646G1-IZS AcuSense 4MP IR Varifocal Bullet Network Camera has been installed
for the primary purpose of using the acquired footage to compute surface velocities using image velocimetry techniques. The
camera has a varifocal lens, which was adjusted to optimise the field-of-view captured by the camera. Camera calibration was
therefore undertaken at the site following installation. The camera captures a $10 \, \mathrm{s}$ video at a frame rate of $20 \, \mathrm{Hz}$ and resolution
of 2688 x 1520 pixels every $15 \, \mathrm{min}$. Ground control points and river cross-sections have been surveyed using a Riegl VZ4000
terrestrial laser scanner and Leica GS14 GPS. Between $30^{\mathrm{th}}$ August and $2^{\mathrm{nd}}$ September 2019, a high-flow event occurred on
the River Feshie and the footage acquired from the fixed camera is used here to illustrate the functionality of KLT-IV. The
processing work-flow for this footage acquired from a fixed monitoring station follows the 'Stationary: GCPs' approach.

In addition to the fixed camera footage, a DJI Phantom 4 Pro UAS was flown at an elevation of approximately $20 \, \mathrm{m}$ above the
water surface during the rising limb of the hydrograph and at the peak river stage. These videos were acquired at a resolution
of 4096 x 2160 pixels at a frame rate of 29.97 fps. The footage was acquired with the camera at $21 - 31°$ from nadir and video
durations of between $30 \, \mathrm{s}$ and $2 \, \mathrm{min}$ are selected for analysis. Two processing options could be considered for the specific
site/flight characteristics: (i) Dynamic: GCPs + Stabilisation; or (ii) Dynamic: GCPs. In using (i), image stabilisation would
first be carried out before orthorectification, under the assumption that stabilisation results in a consistent image sequence,
whereas in (ii): ground control points would be identified and tracked throughout the image sequence, enabling platform
movement to be accounted for. The main limitation for option (i) is that the banks of the channel are heavily vegetated with
variations in elevations of up to $10 \, \mathrm{m}$. The use of features at different elevations for stabilisation negates the assumption of
a planar perspective (Dale et al., 2005), and features may appear to move at different rates, or directions, and this may be
enhanced by the off-nadir perspective of the camera (Schowengerdt, 2006). However, in the case of (ii): GCPs are clearly
visible, and distinctive, across both sides of the channel for the duration of the video. Therefore, these GCPs may be selected
and automatically tracked throughout the image sequence. This information would then be used to automatically correct the
displacement of features on the water surface for movement of the UAS platform. For the reasons outlined above, option (ii)
was chosen for analysis of this case study.

### 2.3.2 Case Study 2: River Coquet, England

The middle reaches of the River Coquet at Holystone in the north-east of England, UK are located $25 \, \mathrm{km}$ downstream from
the river's source in the Cheviot Hills, draining a catchment area of $225 \, \mathrm{km}^2$. This is a wandering gravel-bed river with a





well-documented history of lateral instability (Charlton et al., 2003). On 22<sup>nd</sup> March 2020, during a period of low-flow, a DJI Phantom 4 Pro UAS undertook a flight to acquire imagery along the long-profile of the river. The video footage was acquired at a resolution of 2720 x 1530 pixels and a frame rate of 29.97 fps. Prior to the flight, an Emlid Reach RS+ GPS module was set-up nearby to obtain baseline GPS data, and the UAS was equipped with an Emlid M+ GPS sensor. Both the base and

UAS-mounted GPS acquired L1 GPS data at $14\,\mathrm{Hz}$ and the base station data was used to correct the UAS-mounted GPS logs (i.e. providing a Post-Processed Kinematic (PPK) solution). This enabled the precise position of the UAS to be determined throughout the flight. Taking advantage of this approach, the platform was used to traverse the river corridor at a height of $46\,\mathrm{m}$ above the water surface. The way-points of the pre-planned route were uploaded to the drone using flight management software and the route automatically flown at a speed of $5\,\mathrm{km\,h^{-1}}$. Given the GPS sampling rate and flight speed, the UAS location was

logged ten times for every $1\,\mathrm{m}$ travelled. Synchronisation between the video and GPS was ensured through the mounting of additional hardware. Each time a video begins recording on the DJI Phantom 4 Pro, the front LEDs blink and this was detected using a phototransistor. This event was then logged by the GPS providing a known time that recording began. Timing offsets are also accounted for in this process. Following the flight, inertial measurement unit (IMU) data was downloaded from the UAS. In the case of the DJI Phantom 4 Pro, this is logged at $30\,\mathrm{Hz}$ and is used to determine the camera orientation during the

flight. The process is based upon the assumption that the camera is focussed at nadir, and that the camera gimbal accounts for deviation in the platforms pitch.

## 3   Results

### 3.1   Case Study 1: River Feshie, Scotland

Upon analysis of ten videos acquired from the fixed camera, and four videos acquired from the UAS, flows are reconstructed

for a river stage ranging from $0.785$–$1.762\,\mathrm{m}$, on both the rising and falling-limb of the hydrograph. Analysis of the footage acquired from the UAS and fixed camera enable the generation of a well-defined rating curve relating river stage to flow, with deviations between reconstructed discharge of 4% and 1% in the case of a river stage of $1.762$ and $1.518\,\mathrm{m}$ respectively.

Analysis of each UAS video generated over one-million within-channel trajectories, of which, 20% are shown in the examples within Figure 5. Velocity magnitudes of individual trajectories are presented using a linear colormap with points falling

outside of the lower 99<sup>th</sup> percentile being plotted in black. The plots in Figure 5 are examples of KLT-IV outputs for videos acquired at the peak stage in the observed high flow event on 30<sup>th</sup> August. At this time, peak surface velocities approximated $4\,\mathrm{m\,s^{-1}}$ across the central portion of the channel, decreasing asymmetrically towards the banks. Figure 5 A–B represent the outputs generated from the UAS, whereas C-D represent those from the fixed camera. Due to the vegetated channel boundaries, the UAS was unable to image the flow on the right bank, resulting in approximately $6.5\,\mathrm{m}$ of the water surface requiring ex-

trapolation. However, the main body of flow was successfully captured with no interpolation required. The cubic extrapolation replicates the cross-section flow dynamics well, resulting in just a slight step between the observed and predicted velocities. The computed discharge using the UAS was $82.11\,\mathrm{m^3\,s^{-1}}$ at a stage of $1.762\,\mathrm{m}$.

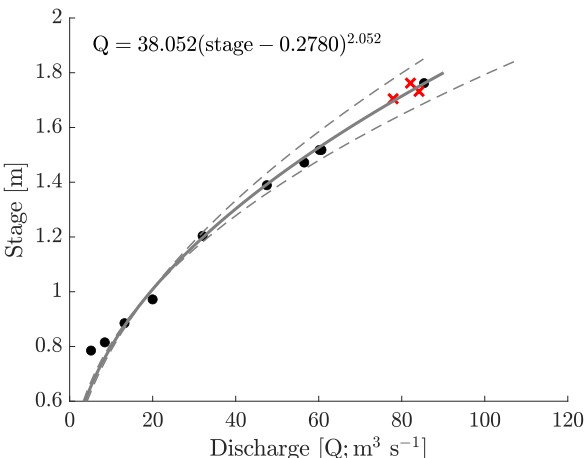

**Figure 4.** Stage-discharge rating curve developed for the River Feshie following image velocimetry analysis using KLT-IV v1.0. The rating curve (grey solid line) is an empirical function with least-squares optimisation of two-parameters with the value of 0.2780 representing the the stage of zero flow. The dashed lines represent the 95% confidence intervals of the rating curve coefficients. River discharge observations produced using the fixed camera are indicated by black circles, whereas the UAS-derived observations are indicated by red crosses.

At the same time, the fixed camera recorded a $10\,\mathrm{s}$ video with the results illustrated in Figure 5 C–D. In contrast to the one-million within-channel trajectories obtained using the UAS (over 60-seconds), 7433 within-channel trajectories were re-
constructed, with the vast majority being detected in the central, fastest flowing part of the channel. As a result of the reduced number of trajectories, some interpolation, and extrapolation to both banks is required. However, the cubic function again clearly replicates the general flow distribution. The lack of trajectories obtained on the left bank may be caused by the camera poorly resolving the features in the near-field, proximal of the camera. Conversely, at the far (right) bank, there are few detectable features in the video and the ground sampling distance of the camera pixels will be relatively low. The peak surface
velocities are in excess of $4\,\mathrm{m\,s^{-1}}$, which are converted to a maximum depth averaged velocity of approximately $3.5\,\mathrm{m\,s^{-1}}$. Using the fixed camera, the computed discharge was $85.38\,\mathrm{m^3\,s^{-1}}$ at a stage of $1.762\,\mathrm{m}$.

### 3.2 Case Study 2: River Coquet, England

A $125\,\mathrm{s}$ flight of the River Coquet generated 3746 sequential images spanning a long-profile distance of $180\,\mathrm{m}$. Upon orthorectification of the imagery using Emlid M+ GPS and UAS IMU data, image sequences were coarsely aligned. To further reduce
the registration error of the imagery, frame-to-frame stabilisation was employed. Following this, the root-mean square error of the ground control point locations was 3.25 pixels (i.e. $6.5\,\mathrm{cm}$). For GCPs one, two, and four, the median distance between the location of individual GCPs relative to the central position of the GCP ($U_{EN}$) is below 2 pixels, with the highest median error reported for GCP three (4.5 pixels) (Figure 6). The interquartile range across GCPs is broadly stable, being less than 2.3 pixels for all except GCP three, which has an interquartile range of 5.7 pixels. Individual GCPs are kept within the field-of-view for



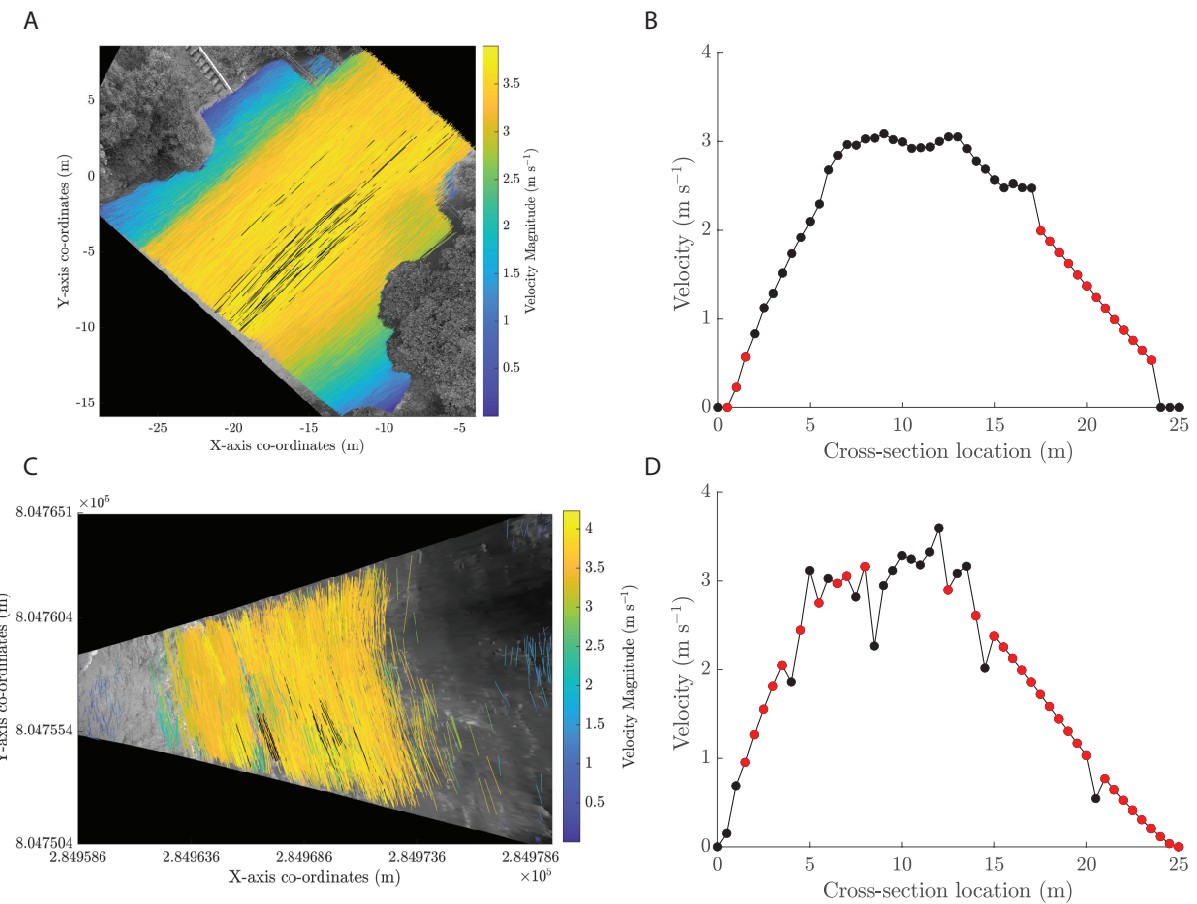

**Figure 5.** KLT-IV example outputs for the River Feshie at a river stage of $1.762\,\mathrm{m}$ using a DJI Phantom 4 Pro Unmanned Aerial System (UAS) (A–B), and a fixed HikVision camera (C–D). The reconstructed discharge was $82.11\,\mathrm{m^3\,s^{-1}}$ and $85.38\,\mathrm{m^3\,s^{-1}}$ for the UAS and fixed platform respectively. Figures A and C illustrate the trajectories and displacement rates of objects tracked on the river surface. Features were tracked for a period of $1\,\mathrm{s}$ and $0.5\,\mathrm{s}$ for A and C respectively. Figures B and D illustrate the depth-averaged velocity for the river cross-section. Black points indicate observations whereas red points indicate nodes of interpolation/extrapolation.



a minimum of $26\,\mathrm{s}$ through to a maximum of $60\,\mathrm{s}$. These findings indicate that the pixel locations of the GCPs are generally stable over time, and that the reconstruction is geometrically consistent i.e. features that appear in a certain location appear in the same location in all predictions where they are in view (Luo et al., 2015).

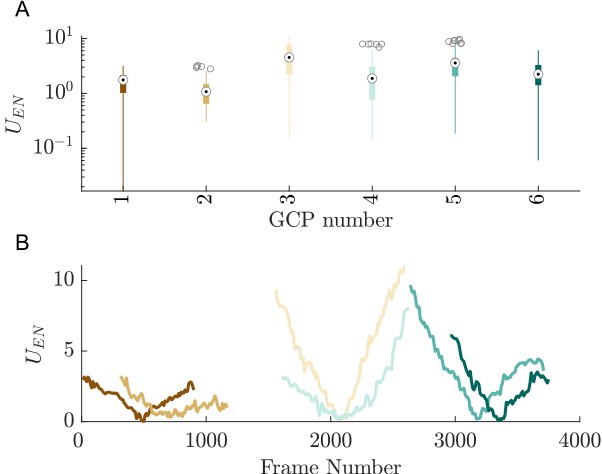

**Figure 6.** $U_{EN}$ indicates how the location of individual GCPs varies relative to the central position of the GCP throughout the stabilised image sequence. Time-averaged results are presented (A) in the form of box-plots with $U_{EN}$ indicating the distance (px) of GCPs in individual frames relative to the central position of the GCP. The location of GCPs were manually determined for every $10^{\mathrm{th}}$ frame in the stabilised image sequence. The variation of the GCP locations over time, relative to the central position, is provided in (B). Line colors are consistent with the GCP numbers provided in (A).

This stabilised imagery was subsequently used for image velocimetry analysis. This yielded a total of 19 million tracked features, of which, 5% are displayed in Figure 7. Velocity magnitudes of individual trajectories are presented using a linear

colormap with points being displayed if they lie within the lower $99.99^{\mathrm{th}}$ percentile. The vast majority of tracked features exhibit negligible apparent movement, with a median displacement of $0.01\,\mathrm{m\,s}^{-1}$, as would be expected given the significant areas of vegetated surfaces imaged by the UAS. The interquartile range of measurements spans $0.008$ - $0.02\,\mathrm{m\,s}^{-1}$. The data are positively skewed ($s = 10.4$) as a result of the majority of the identified features representing static areas of the landscape (i.e. features beyond the extent of the active channel), with the long-tail of the distribution representing areas of motion.

25% of tracked features exhibit a velocity in excess of $0.2\,\mathrm{m\,s}^{-1}$. These are predominantly located within the active channel margins, although a cluster of points is also evident to the lower-left corner of Figure 7. Whereas trajectories plotted within the main channel represents detected motion of the water surface, the movement to the lower-left represents apparent motion of vegetation. These elements of the landscape were not used in the stabilisation process due to the treetops being at a significantly different elevation from the river channel. This apparent motion therefore illustrates the way in which parts of the image of

different elevations can generate differential displacement rates (as discussed in Section 2.3.1). Maximum velocities within the main channel approximate $1\,\mathrm{m\,s}^{-1}$ towards the lower extent of the field-of-view. Within this part of the river reach, the active width narrows and depth does not appreciably increase, therefore resulting in the localised increase in velocity magnitude.



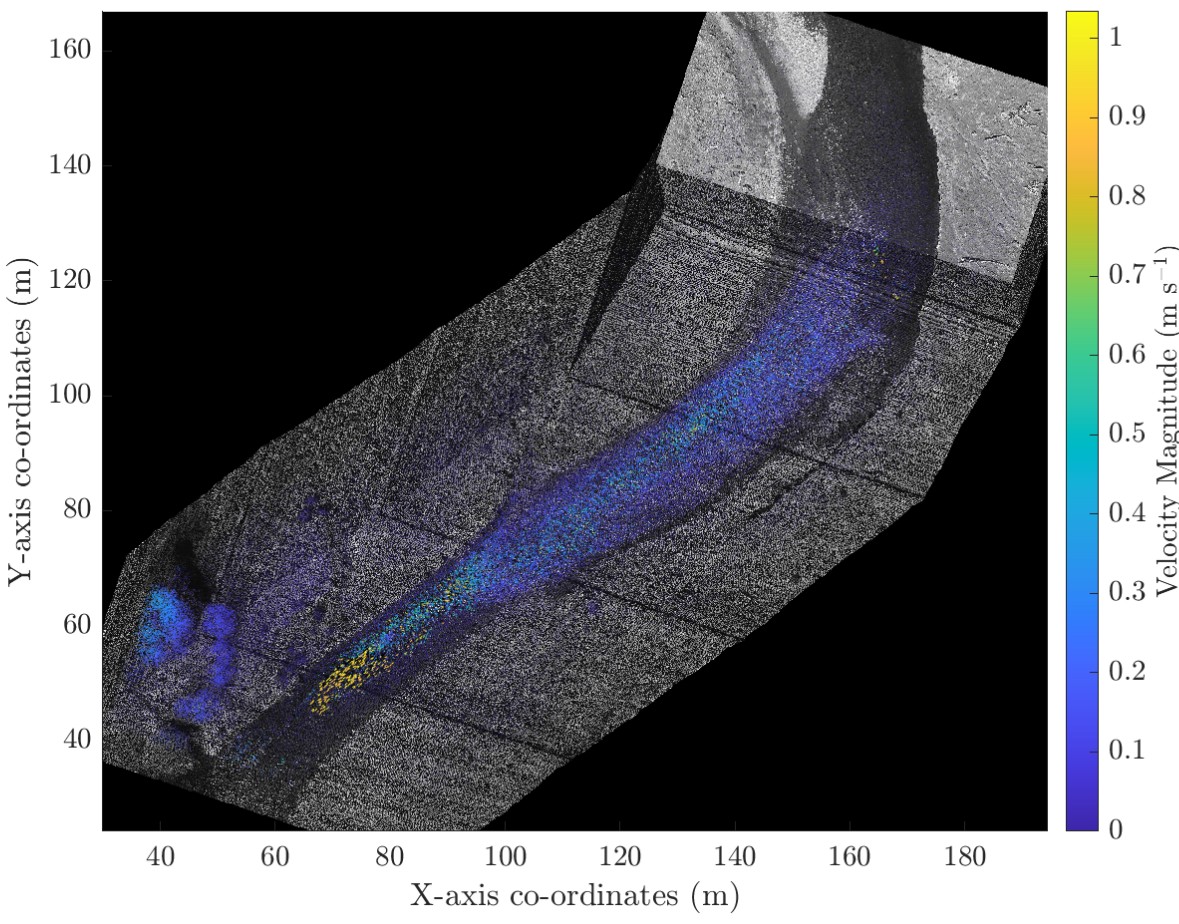

**Figure 7.** KLT-IV v1.0 outputs illustrating the apparent velocities of features within the river corridor of the River Coquet (UK) following analysis using differential GPS and IMU data to orthorectify the imagery prior to stabilisation and image velocimetry analysis.

## 4 Discussion

KLT-IV offers a flexible, PTV-based approach for the determination of river flow velocity and river discharge across a range of
hydrological conditions. The software offers the user a range of options that may be chosen depending on the site conditions, environmental factors at the time of acquisition, and the desired outputs. Platform movement can be accounted for through the use of either ground control points, or features that are stable within the field of view. These approaches are consistent with work-flows provided in other image velocimetry software packages. However, additional features are also provided. KLT-IV offers the user the opportunity to determine river flow velocities without the presence of ground control points. For example,
under the assumption that the camera is at nadir, the camera model is selected, and the sensor height above the water surface is known, flow velocities can be determined. This has the potential to be used for opportunist flow gauging from a bridge, or using UAS platforms where the platform is stable, camera is at nadir, and ground control points are not visible. This may be





particularly useful for wide river cross-sections, or where surveying of ground control points is problematic. It is also possible for work-flows to be combined. For example, in the situation where UAS-based footage has been acquired at nadir, an initial
analysis could be achieved using the 'Stationary: Nadir' approach, which would provide an estimate under the assumption that the platform is stable. If, however, movement in the platform does occur, the orthorectified footage could be subsequently analysed using the 'Dynamic: Stabilisation' approach to address any movement in the platform. These approaches have the potential to streamline the data acquisition procedure for image velocimetry analysis under conditions where image velocimetry measurements may be problematic (e.g. during flood flow conditions where site access is limited).

Within KLT-IV, a novel approach of integrating external sensors (namely GPS and IMU data) has the potential to extend the adoption of image velocimetry approaches beyond the local scale, and enable reach scale variations in hydraulic processes to be examined. Navigating a UAS platform for several hundreds of meters (and potentially km's) for the purposes of acquiring distributed, longitudinal velocity measurements have several applications including the mapping and monitoring of physical habitats (Maddock, 1999), for the calibration and validation of hydraulic models in extreme floods, and quantification of forces
driving morphological adjustment (e.g. bank erosion). However, this approach does require further testing and validation. As outlined by Huang et al. (2018), the reliance on sensors to determine the 3-dimensional position of an UAS platform at the instance when measurements (e.g. images) are acquired can be affected by the time offset between instruments, the quality of the differential GPS data, the accuracy of the IMU and ability of the camera gimbal to account for platform tilt and roll. However, when using an UAS for image velocimetry analysis, the requirement of low-speeds (e.g. $5\,\mathrm{km\,h^{-1}}$) will diminish the influence
of timing discrepancies (e.g. between camera trigger and GPS measurement) on positional errors. For example, an unaccounted time offset of $15\,\mathrm{ms}$ would equate to a positional error of $0.021\,\mathrm{m}$ assuming a flight speed of $5\,\mathrm{km\,h^{-1}}$, a value within the tolerances of most differential GPS systems. More precise orthophotos could be generated using IMU and GPS devices with higher sensitivity but this would come at increased cost (Bandini et al., 2020). To overcome potential hardware limitations, the proposed GPS + IMU work-flow utilises stable features within the camera's field-of-view to account for positional errors and
this has resulted in the generation of geometrically consistent image sequences for use within an image velocimetry work-flow (Figure 7). However, the transferability of this approach should be the subject of further research and testing across a range of conditions e.g. higher-density and diversity of vegetation cover. In instances where the GPS + IMU data alone (i.e. without stabilisation) produces sufficiently accurate orthophotos, the generated orthophotos may be subsequently used with the 'Dynamic: Stabilisation' orientation which would eliminate the need for the stabilisation routine and the requirement of the user
identifying stable features within the frame sequence. Finally, this approach operates under the assumption that the distance between the camera and water surface is consistent throughout the footage. Whilst this approximation may hold for relatively short sections, or river reaches with shallow gradients, this may become problematic when surface slope is considerable, and/or the surveyed reach is sufficiently long.

Whilst KLT-IV offers several new features for image velocimetry analysis, further development of the software is planned
which will: (i) embed a suite of image pre-processing methods; (ii) enable post-processing (filtering) of the feature trajectories to eliminate spurious velocity vectors; (iii) provide additional feature detection approaches (e.g. FAST, SIFT) in order to provide flexibility; and (iv) provide the option of analysing multiple videos (e.g. from a fixed monitoring station) to facilitate



the generation of time-series of river flow observations. This processing will be possible using the user's local machine, and also enable the user to transfer footage to the Newcastle University High Performance Computing cluster and file sharing
system for remote analysis.

## 5 Conclusions

KLT-IV v1.0 software provides an easy-to-use graphical interface for sensing flow velocities and determining river discharge in river systems. The basis for the determination of flow rates is the implementation of a novel PTV-based approach to tracking visible features on the water surface. Velocities can be determined using either mobile camera platforms (e.g. UAS) or fixed
monitoring stations. Camera motion and scaling from pixel to real-world distances is accounted for using either ground control points, stable features within the field-of-view, or external sensors (consisting of differential GPS and inertial measurement unit data). Conversely, if the platform is stable, scaling from pixel to real-world distances may be achieved through the use of either ground control points, or by defining the known distance between the camera and the water surface (when the camera model is known and view at nadir). This flexibility offers the user with a range of options depending on the mode of data acquisition.
To illustrate the use of KLT-IV two case studies from the UK are presented. In the first case study, footage is acquired from a UAS and fixed camera over the duration of a high-flow event. Using this footage, a well-defined flow rating curve is developed with deviations between the fixed and UAS-based discharge measurements in the order of $< 4\%$. In the second case study, a UAS is deployed to acquire footage along a $180\,\mathrm{m}$ reach of river. Equipped with a differential GPS sensor and travelling at a speed of $5\,\mathrm{km\,h^{-1}}$, video footage acquired over a period of $125\,\mathrm{s}$ is used to successfully reconstruct surface velocities along
the river reach without the use of ground control points. These examples are provided to illustrate the potential for KLT-IV to be used for quantifying flow rates using videos collected from fixed, or mobile camera systems.

## 6 Software Availability

KLT-IV v1.0 is freely available to download from: https://sourceforge.net/projects/klt-iv/. During the installation of KLT-IV v1.0 an active internet connection is required as the MATLAB 2019b Runtime will be downloaded and installed, if not already
present on the operating system. Datasets used in the production of this article, along with the settings adopted within KLT-IV v.1.0 can be downloaded at: https://zenodo.org/record/3882254#.XuCwcUVKj-g. The Digital Object Identifier (DOI) of the dataset is: 10.5281/zenodo.3882254. A Google Group has been established for the community of users to pose questions and comments about the software at: https://groups.google.com/forum/#!forum/klt-iv-image-velocimetry-software.

## 7 Hardware Requirements

The software can run on any of the following operating systems: Windows 10 (version 1709 or higher), Windows 7 Service Pack 1, Windows Server 2019, Windows Server 2016. The minimum processor requirement is any Intel or AMD x86-64 processor. However, it is recommended that the processor has four logical cores and AVX2 instruction set support. At least 3





GB of HDD space is required. Minimum memory requirements are 4 GB, but 8 GB is recommended. No specific graphics card is required, however, a hardware accelerated graphics card supporting OpenGL 3.3 with 1GB GPU memory is recommended.

*Competing interests.* The author declares no conflict of interest.

*Acknowledgements.* This project was instigated when the author was employed on the UK Research and Innovation NERC-funded project: 'Susceptibility of catchments to INTense RAinfall and flooding (SINATRA)'. Funding of article processing charges has been provided by NE/K008781/1. The author thanks Professor Andy Russell and Professor Andy Large for support in developing this software, to Sophie Pearce for testing and providing feedback on the software, and to the attendees of a workshop at Newcastle University in Summer 2019, which
informed software development. The splash screen, and icon for KLT-IV v1.0 was produced by Kelly Stanford (https://www.kellystanford. co.uk).





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





**Appendix A**

List of acceptable video file formats:

.asf - ASF File

610    .asx - ASX File

.avi - AVI File

.m4v - MPEG-4 Video

.mj2 - Motion JPEG2000

.mov - QuickTime movie

615    .mp4 - MPEG-4

.mpg - MPEG-1

.wmv - Windows Media Video