# Peer review of "KLT-IV v1.0: Image velocimetry software for use with fixed and mobile platforms"

_Geoscientific Model Development, 2020_

## Referee Comment (RC1) · Frank Engel (Referee) · 6 Aug 2020

**1   General comments**

The author reports on a new software for computing river surface velocity and discharge from the use of video captured by fixed or mobile platforms, including webcameras installed at river gauges, and UAS. The software, KLT-IV v1.0, presents a complete processing package that would enable users to go from raw video to discharge results. KLT-IV uses a combination of feature tracking algorithms (in this case Good Feature to Track) and Optical Flow to compute trajectories of the objects of interest. Among other novel aspects of the software, this approach allows not just only for the tracking of water surface velocity features, but also for ground control features.

By incorporating this tracking functionality, the author has created a software package that can enable some new approaches to managing scene and camera orthorectification. In my opinion, this is an excellent addition to the growing suite of surface velocity tools which have appeared in the scientific literature over the past 5 or so years. The potential is there with KLT-IV to begin to standardize reach-based UAS surface velocity surveys, and yet the software also provides the necessary functions for standard fixed or mobile platform camera gaging. Well done.

This paper is well organized and coherent. It clearly states the aims of the work, and the author adequately anchors this work into the body of literature. The functionality and workflow of the KLT-IV software is clearly presented. The style and clarity of prose is excellent. Overall, this is an excellent paper that is nearly ready for publication.

**2 Specific comments**

I would like to see some more discussion included in the paper about how well the KLT-IV flow trajectory algorithms perform compared to other algorithms and independent measurement techniques. At the least, a little discussion of the results from the cited work by Pearce et al. (2020) would be well received. Has the author collected independent flow velocity and/or discharge measurements and compared them with the output from KLT-IV since the seminal technical note published in 2016? It would be very good to address any new findings here, even if only briefly, or by citing associated literature.

I would also like to see some text added in the discussion indicating known and common method failure points (more generically, rather than just specifically associated with the two case studies presented). What are the common minimum seeding or velocity thresholds in which the method begins to struggle? Are there strategies on balancing the input/processing frame rate and anticipated flow velocity? Any guidance

or insights on these factors may help ensure the KLT-IV software is used for its intended purpose, and that results are as accurate as possible.

Finally, I would like to see some information about the processing times and expectations for compute hours for use of the KLT-IV software under certain conditions. What computer hardware was used to compute the case study results? What sort of processing time did it take to do these case studies? Have any formal bench testing experiments been undertaken (in addition to the work by Pearce et al., 2020)? Although the hardware requirements section addresses the basic needs in order to run the software, should a user plan to use cluster computers for more extensive use of KLT-IV? What about the ability to port the software to operate on edge computing devices? Perhaps, if not at least mentioned in this paper, there may be a reason to write another paper discussing these things.

**3  Technical corrections**

Overall, this paper is well constructed, and highly relevant to the field of non-contact remote sensing of hydrometric variables. I would have no hesitation approving this manuscript after my comments are addressed. In addition to my points above, a few minor issues are discussed below, referenced to the line numbers as indicated in the pre-print version of the manuscript.

Line 33: The Despax et al. (2019) paper was really about determining the interlaboratory uncertainty between how we do direct streamflow measurements with ADCP. I wouldn't necessarily say it is about remotely operated streamflow monitoring, as is implied by line 31.

Line 60: It is my hope that soon, we will be able to capture topographic and bathymetric observations at the same time, in a non-contact fashion, as we capture surface velocities with image velocimetry techniques. Much promise and development seems

to be happening now with the use of tuned, multi-phased ground penetrating radar to capture the channel bottom characteristics (by drone or cable way). This is an exciting time for non-contact hydraulic remote sensing.

Line 120: You can also cite RIVeR here as well. The RIVeR typical workflow rectifies the results from PIV conducted on non-transformed image pairs.

- Patalano, García, and Rodríguez, "Rectification of Image Velocity Results (RIVeR): A Simple and User-Friendly Toolbox for Large Scale Water Surface Particle Image Velocimetry (PIV) and Particle Tracking Velocimetry (PTV)." 10.1016/j.cageo.2017.07.009

Line 171: Does this imply that if a UAS or fixed image scene with excessive motion is not completely corrected, the error detection result would censor data which may be valid? Or, in a more positive view, censor data which still show motion contamination?

Line 207: Any particular reason why the camera positions inputs are required as radians, rather than degrees? Use of atan2 in the conversion process within KLT-IV would easily handle any typical issues that arise from converting from a world geometry convention (degrees) to a polar geometry convention (radians), and would be much simpler for the end user.

Line 310: Please either define that mAOD is Ordinance Datum, or consider converting to some other widely recognized reference. Your international readers may not be familiar with mAOD.

Line 445: A useful point here could be made for UAS terrain-following flight planning. This functionality is capable with more sophisticated ground control stations, such as Mission Planner. Moreover, some of the newer consumer-grade UAS on the market now are beginning to incorporate Terrain-following functionality.

---

## Referee Comment (RC2) · Anonymous Referee #2 · 9 Aug 2020

General comments

The author presents a new software for the determination of river surface flow velocities and river discharge using videos. It is based on a combination of optical flow and automated corner point detection algorithms. The underlaying detection algorithm used is the Good Features To Track and the tracking is done by using the Kanade Lucas Tomasi method.

The software is freely available, it has a clean and intuitive graphical interface. It has a very good set of options for camera calibration / stabilization. To my knowledge there is no freely available software which uses the same method, hence it is a good addition to the already available tools. The author has also done a good work on keeping the amount of parameters to a minimum and giving default values for them.

[Figure]

The paper is clearly written, with a good description of the software functionalities.

Specific comments

My main comments are related to the validation and limitations of the software and of the algorithms implemented in it.

The author presents two case studies, the first one, the River Feshie where 10 videos taken from a fix camera and 4 videos from UAS were processed. The results are used to fit a rating curve, the deviations between the reconstructed rating curve and the measurements is mentioned to be 4%. However, it would be desirable to have a comparison against a different methodology e.g. ADCP. Has the author performed such comparison?

In Figure 4 it can be observed that the measured discharges deviate the most from the rating curve at low flows. It seems that the implemented method gets less accurate results for lower velocities. This brings me to my second comment. The paper is missing a section where the software limitations are explained, for example what are the minimum velocities? Are there a minimum set of characteristics to be fulfilled, e.g shadows, glare, type of flow, minimum camera angle, minimum video duration, etc.?

The results from processing a video recorded with a fixed camera and with an UAV at the same river stage are shown in Figure 5. The trajectories are qualitatively different, what is the reason for that? Is it because of the angle of view of the fix camera? Is it related to the orthorectification process? What are the limitations?

It would also be nice to see some insights on the uncertainty of the model and sensitivity of the parameters. This would help to chose the right value for them.

Technical corrections

In line 140 it is mentioned that the free-surface image velocity measurements must be translated into a depth-averaged velocity, however it is never explicitly mentioned that the Alpha value in the GUI is meant for that.

Line 188. It is mentioned that the mode 'Single video' is the default one, but there no other modes. This should be mentioned here or, if possible, change this field in the GUI until another mode is implemented.

Lines 410-455 (discussion section) I think it would be better to focus this section on the limitations and accuracy of the software, or to add that to the discussion.

Line 479. Add the word "software"

I tried to ran some cases but I could only process one: /Feshie/FixedCam/Video_02, for all the other cases that I tried, the software crashed, without much information about the source of the crash. For the case that I was able to process, I got a value which was out of the reconstructed rating curve, probably one of the provided files is not correct.

---

## Author Comment (AC1) · 30 Sep 2020

**Response to Reviewer 1 (Dr Frank Engel)**

**General Comments**

**Reviewer Point P 1.1** — The author reports on a new software for computing river surface velocity and discharge from the use of video captured by fixed or mobile platforms, including webcameras installed at river gauges, and UAS. The software, KLT-IV v1.0, presents a complete processing package that would enable users to go from raw video to discharge results. KLT-IV uses a combination of feature tracking algorithms (in this case Good Feature to Track) and Optical Flow to compute trajectories of the objects of interest. Among other novel aspects of the software, this approach allows not just only for the tracking of water surface velocity features, but also for ground control features. By incorporating this tracking functionality, the author has created a software package that can enable some new approaches to managing scene and camera orthorectification. In my opinion, this is an excellent addition to the growing suite of surface velocity tools which have appeared in the scientific literature over the past 5 or so years. The potential is there with KLT-IV to begin to standardize reach-based UAS surface velocity surveys, and yet the software also provides the necessary functions for standard fixed or mobile platform camera gaging. Well done. This paper is well organized and coherent. It clearly states the aims of the work, and the author adequately anchors this work into the body of literature. The functionality and workflow of the KLT-IV software is clearly presented. The style and clarity of prose is excellent. Overall, this is an excellent paper that is nearly ready for publication.

**Reply**: I would like to extend my thanks to Dr Engel for the detailed comments and suggestions made within the review provided. In this document I will respond to each comment individually, and outline the changes that are made in the revised submission.

**Specific Comments**

**Reviewer Point P 1.2** — I would like to see some more discussion included in the paper about how well the KLT-IV flow trajectory algorithms perform compared to other algorithms and independent measurement techniques. At the least, a little discussion of the results from the cited work by Pearce et al. (2020) would be well received. Has the author collected independent flow velocity and/or discharge measurements and compared them with the output from KLT-IV since the seminal technical note published in 2016? It would be very good to address any new findings here, even if only briefly, or by citing associated literature.

**Reply**: At the locations of the two case studies presented in this article, I have not been able to acquire velocity measurements using standard methods whilst also capturing footage for image velocimetry analysis. Within the Method section (Lines 163–165) and in the newly introduced Section 5: 'Challenges and Future Development' (Lines 485–487), the findings presented in the Pearce et al (2020) study are introduced. These are the only published inter-comparisons between KLT-IV and other approaches at this time. This lack of formal assessment will be addressed in further works.

**Reviewer Point P 1.3** — I would also like to see some text added in the discussion indicating known and common method failure points (more generically, rather than just specifically associated with the two case studies presented). What are the common minimum seeding or velocity thresholds in which the method begins to struggle? Are there strategies on balancing the input/processing frame rate and anticipated flow velocity? Any guidance or insights on these factors may help ensure the KLT-IV software is used for its intended purpose, and that results are as accurate as possible.

**Reply**: Guidance about the key limitations of KLT-IV software have been highlighted in Section 5: 'Challenges and Future Development'. Here I present guidance related to the minimum required image resolution, requirements related to the presence and distribution of features to track, considerations relating to image illumination, and I also note a key limitation of the software, namely the lack of post-processing options for filtering spurious trajectories. Presented in Appendix C is a sensitivity analysis to determine how the software's performance varies with changes to the user-defined 'frame extract rate' and the 'block size'. This is also discussed in Lines 396–402. An objective of future works will be to assess the sensitivity of the software to varying levels of seeding densities, particle clustering, image illumination, etc. across a range of flow conditions.

**Reviewer Point P 1.4** — Finally, I would like to see some information about the processing times and expectations for compute hours for use of the KLT-IV software under certain conditions. What computer hardware was used to compute the case study results? What sort of processing time did it take to do these case studies? Have any formal bench testing experiments been undertaken (in addition to the work by Pearce et al., 2020)? Although the hardware requirements section addresses the basic needs in order to run the software, should a user plan to use cluster computers for more extensive use of KLT-IV? What about the ability to port the software to operate on edge computing devices? Perhaps, if not at least mentioned in this paper, there may be a reason to write another paper discussing these things.

**Reply**: Within Appendix B I now provide a Table which documents the processing times, and memory utilisation for each of the videos presented in this article. This is also referred to in Section 2.3 (Lines 312–313).

The current version of the software can only be used on PCs running Windows operating systems, and is also limited by the processor unit (e.g. it will not run on ARM CPUs). Whilst beyond the scope of this initial release, in future releases I am hopeful of being able to provide support for edge processing on devices where the KLT-IV application does not run (e.g. on a Raspberry Pi in conjunction with MATLAB Online), in addition to batch processing using the Newcastle University high performance computing (HPC) service.
* * *
**Technical Corrections**

**Reviewer Point P 1.5** — Line 33: The Despax et al. (2019) paper was really about determining the interlaboratory uncertainty between how we do direct streamflow measurements with ADCP. I wouldn't necessarily say it is about remotely operated streamflow monitoring, as is implied by line 31.

**Reply**: This reference has been replaced with a more appropriate one: Le Coz, J., Pierrefeu, G., Paquier, A. (2008) Evaluation of river discharges monitored by a fixed side-looking Doppler profiler. Water Resources Research, 44(4), 10.1029/2008WR006967.

**Reviewer Point P 1.6** — Line 60: It is my hope that soon, we will be able to capture topographic and bathymetric observations at the same time, in a non-contact fashion, as we capture surface velocities with image velocimetry techniques. Much promise and development seems to be happening now with the use of tuned, multi-phased ground penetrating radar to capture the channel bottom characteristics (by drone or cable way). This is an exciting time for non-contact hydraulic remote sensing.

**Reply**: I share your optimism here and look forward to seeing how these technologies can be fused.

**Reviewer Point P 1.7** — Line 120: You can also cite RIVeR here as well. The RIVeR typical workflow rectifies the results from PIV conducted on non-transformed image pairs: Patalano, García, and Rodríguez, "Rectification of Image Velocity Results (RIVeR): A Simple and User-Friendly Toolbox for Large Scale Water Surface Particle Image Velocimetry (PIV) and Particle Tracking Velocimetry (PTV)", 10.1016/j.cageo.2017.07.009

**Reply**: This reference has now been added to the manuscript at this location.

**Reviewer Point P 1.8** — Line 171: Does this imply that if a UAS or fixed image scene with excessive motion is not completely corrected, the error detection result would censor data which may be valid? Or, in a more positive view, censor data which still show motion contamination?

**Reply**: The error detection should not be affected by residual motion after stabilisation as the forward tracking and backward tracking are based on the same image sequence, albeit in reverse order. Features would only be removed from the analysis if during the backward propagation the location of a tracked feature appears to differ from the initial solution.

**Reviewer Point P 1.9** — Line 207: Any particular reason why the camera positions inputs are required as radians, rather than degrees? Use of atan2 in the conversion process within KLT-IV would easily handle any typical issues that arise from converting from a world geometry convention (degrees) to a polar geometry convention (radians), and would be much simpler for the end user.

**Reply**: There is no good reason for the input being in radians rather than degrees and you make a good point about degrees being more user friendly. This change will be implemented in the next release of the software.

**Reviewer Point P 1.10** — Line 310: Please either define that mAOD is Ordinance Datum, or consider converting to some other widely recognized reference. Your international readers may not be familiar with mAOD.

**Reply**: This sentence has now been reworked to read: 'The headwaters originate in the Cairngorm National Park at an elevation of 1263m above the Newlyn Ordnance Datum (AOD) before joining the River Spey at an elevation of 220mAOD.' (Lines 315–318).

**Reviewer Point P 1.11** — Line 445: A useful point here could be made for UAS terrain-following flight planning. This functionality is capable with more sophisticated ground control stations, such as Mission Planner. Moreover, some of the newer consumer-grade UAS on the market now are beginning to incorporate Terrain-following functionality.

**Reply**: Thanks for this suggestion. This option has been incorporated into the text at Lines 469–474, which reads: 'An alternative solution for ensuring that the distance between the UAS and water surface remains constant over time may be to use flight planning software (e.g. fly litchi mission planner). This would enable the user to define the altitude of the flight above the earth surface (as defined by a digital elevation model), rather than above the elevation at take-off. However, in this instance, the GPS log would need to be modified to ensure the recorded GPS height was constant and that this value minus the specified WSE corresponds with the known flight height above the water surface'.

---

## Author Comment (AC2) · 30 Sep 2020

**General Comments**

**Reviewer Point P 2.1** — The author presents a new software for the determination of river surface flow velocities and river discharge using videos. It is based on a combination of optical flow and automated corner point detection algorithms. The underlaying detection algorithm used is the Good Features To Track and the tracking is done by using the Kanade Lucas Tomasi method. The software is freely available, it has a clean and intuitive graphical interface. It has a very good set of options for camera calibration / stabilization. To my knowledge there is no freely available software which uses the same method, hence it is a good addition to the already available tools. The author has also done a good work on keeping the amount of parameters to a minimum and giving default values for them. The paper is clearly written, with a good description of the software functionalities.

**Reply**: I would like to extend my thanks to the reviewer for the detailed comments and suggestions made, all of which have enabled the improvement of this manuscript. In this document I will respond to each comment individually, and outline the changes made to the revised submission.

**Specific Comments**

**Reviewer Point P 2.2** — My main comments are related to the validation and limitations of the software and of the algorithms implemented in it.

The author presents two case studies, the first one, the River Feshie where 10 videos taken from a fix camera and 4 videos from UAS were processed. The results are used to fit a rating curve, the deviations between the reconstructed rating curve and the measurements is mentioned to be 4%. However, it would be desirable to have a comparison against a different methodology e.g. ADCP. Has the author performed such comparison?

**Reply**: At the locations of the two case studies presented in this article, I have not been able to acquire velocity measurements using standard methods whilst concurrently capturing footage for image velocimetry analysis. Within the Methods sections (Lines 163–166) and in the newly introduced Section 5: 'Challenges and Future Development' (Lines 485–487), findings presented by the Pearce et al (2020) study are introduced. These are the only published inter-comparisons between KLT-IV and other approaches at this time. This lack of formal assessment will be addressed in further works.

**Reviewer Point P 2.3** — In Figure 4 it can be observed that the measured discharges deviate the most from the rating curve at low flows. It seems that the implemented method gets less accurate results for lower velocities. This brings me to my second comment. The paper is missing a section where the software limitations are explained, for example what are the minimum velocities? Are there a minimum set of characteristics to be fulfilled, e.g shadows, glare, type of flow, minimum camera angle, minimum video duration, etc.?

**Reply**: As a complete assessment of KLT-IV's performance at this site relative to standard techniques has yet to be completed, it is difficult at this point to draw conclusions about its performance. This is currently the subject of further assessment and research. I have now added a section which goes into some detail about the limitation of the software. In Section 5: 'Challenges and Future Development' (lines 475–504). I present guidance related to the minimum required image resolution, requirements related to the presence and distribution of features to track, considerations relating to image illumination, and I also note a key limitation of the software, namely the lack of post-processing options for filtering spurious trajectories. An objective of future works will be to assess the sensitivity of the software to varying levels of seeding densities, clustering, image illumination, etc. across a range of flow conditions.

**Reviewer Point P 2.4** — The results from processing a video recorded with a fixed camera and with an UAV at the same river stage are shown in Figure 5. The trajectories are qualitatively different, what is the reason for that? Is it because of the angle of view of the fix camera? Is it related to the orthorectification process? What are the limitations?

**Reply**: The dataset from the fixed camera is certainly noisier than that acquired from the UAS and this is likely to be due to several factors, not least the short duration fixed camera video (10-s) relative to the UAS (60-s). Furthermore, at this site, under the high-flow conditions presented in Figure 5, the water surface deviates from the planar assumption that is required for the analysis. The UAS footage will be less sensitive to these local changes in water surface elevations than the fixed camera. Therefore, the oblique camera angle of the fixed camera is likely to produce less favourable results. This information has been incorporated into the manuscript text at Lines 387–395).

**Reviewer Point P 2.5** — It would also be nice to see some insights on the uncertainty of the model and sensitivity of the parameters. This would help to chose the right value for them.

**Reply**: Analysis of the sensitivity of KLT-IV to the two main user defined parameters has been undertaken for three of the fixed camera videos acquired at the River Feshie. Results are presented in Appendix C, and a discussion of this is presented at lines 396–402. The text reads: 'An illustration of how the generated outputs vary with changes to user-defined settings of extract rate (s) and block size (px) are demonstrated for a selection of the fixed videos acquired at the Feshie monitoring station (Appendix C). Generally, varying these two parameters results in relatively small changes to the velocity profile, with the mean values of the reconstructed velocity profile ranging from  $0.89-0.94 \,\mathrm{m\,s^{-1}}$  (Video 8),  $1.18-1.29 \,\mathrm{m\,s^{-1}}$  (Video 2), and  $1.68-1.80 \,\mathrm{m\,s^{-1}}$  (Video 6). In each of these examples, the selection of a broad range of input settings resulted the cross-sectional average velocity varying by less than 10%. Of note however, is that deviations in the velocity profile are most sensitive to changes in these parameters in the near-field where features may transit the scene rapidly, and the far field where features are difficult to resolve.'

**Technical Corrections**

**Reviewer Point P 2.6** — In line 140 it is mentioned that the free-surface image velocity measurements must be translated into a depth-averaged velocity, however it is never explicitly mentioned that the Alpha value in the GUI is meant for that.

**Reply**: This was accidentally omitted from the original submission and has been added at Lines306–308. The newly inserted text reads: 'Finally, an Alpha value needs to be provided. This is the ratio used to convert the measured surface velocities to depth-averaged velocity, which is then used in the calculation of discharge. A default value of 0.85 is generally appropriate if no supplementary data is available to inform the user (see Section 1.3 for more information).'

**Reviewer Point P 2.7** — Line 188. It is mentioned that the mode 'Single video' is the default one, but there no other modes. This should be mentioned here or, if possible, change this field in the GUI until another mode is implemented.

**Reply**: To ensure that the user interface does not vary too much from version-to-version, the video mode was inserted into v1.0 despite no other alternative. In v1.1 'Multiple videos' will be enabled. The text at Line 190–193 has been modified to read: 'The first section: Video Inputs, is where the video acquisition details are provided. Within v1.0 of the software, only 'Single Video' mode can be selected, meaning that only one video at a time can be analysed, and this video may be selected using the file selection dialog box.'

**Reviewer Point P 2.8** — Lines 410-455 (discussion section) I think it would be better to focus this section on the limitations and accuracy of the software, or to add that to the discussion.

**Reply**: Please see response to P 2.3.

Reviewer Point P 2.9 — Line 479. Add the word "software"

**Reply**: This has been modified to read 'Software and Hardware Requirements' (Line 527).

**Reviewer Point P 2.10** — I tried to ran some cases but I could only process one: /Feshie/ FixedCam/Video\_02, for all the other cases that I tried, the software crashed, without much information about the source of the crash. For the case that I was able to process, I got a value which was out of the reconstructed rating curve, probably one of the provided files is not correct.

**Reply**: I hope to be able to understand the nature of the issues encountered more fully to hopefully resolve this. If possible, please could you provide more information as a comment, or post to the software forum at: https://groups.google.com/forum/#!forum/klt-iv-image-velocimetry-software with more information.